# Segmentation Performance Comparison Considering Regional Characteristics in Chest X-ray Using Deep Learning

**DOI:** 10.3390/s22093143

**Published:** 2022-04-20

**Authors:** Hyo Min Lee, Young Jae Kim, Kwang Gi Kim

**Affiliations:** 1Department of Biomedical Engineering, College of Health Science, Gachon University, Incheon 21936, Korea; hn5597@gachon.ac.kr (H.M.L.); youngjae@gachon.ac.kr (Y.J.K.); 2Department of Health Sciences and Technology, Gachon Advanced Institute for Health Sciences and Technology (GAIHST), Gil Medical Center, Gachon University, Incheon 21936, Korea

**Keywords:** deep learning, segmentation, chest X-ray, properties by areas

## Abstract

Chest radiography is one of the most widely used diagnostic methods in hospitals, but it is difficult to read clearly because several human organ tissues and bones overlap. Therefore, various image processing and rib segmentation methods have been proposed to focus on the desired target. However, it is challenging to segment ribs elaborately using deep learning because they cannot reflect the characteristics of each region. Identifying which region has specific characteristics vulnerable to deep learning is an essential indicator of developing segmentation methods in medical imaging. Therefore, it is necessary to compare the deep learning performance differences based on regional characteristics. This study compares the differences in deep learning performance based on the rib region to verify whether deep learning reflects the characteristics of each part and to demonstrate why this regional performance difference has occurred. We utilized 195 normal chest X-ray datasets with data augmentation for learning and 5-fold cross-validation. To compare segmentation performance, the rib image was divided vertically and horizontally based on the spine, clavicle, heart, and lower organs, which are characteristic indicators of the baseline chest X-ray. Resultingly, we found that the deep learning model showed a 6–7% difference in the segmentation performance depending on the regional characteristics of the rib. We verified that the performance differences in each region cannot be ignored. This study will enable a more precise segmentation of the ribs and the development of practical deep learning algorithms.

## 1. Introduction

A chest X-ray (CXRs) is essential for diagnosing rib fracture, heart shape, blood vessel expansion, pulmonary vessel size, and pulmonary edema [1]. However, X-ray images are difficult to read because several anatomic structures cross and overlap in the chest [2], making it is difficult for radiologists to identify diseases hidden in a chest background where organs overlap [3,4]. Foreground bones such as ribs and clavicles have been reported to cover 82%–95% of undiscovered lung cancers [5].

Thus, chest background structure removal or suppression algorithm is used to observe the chest area more clearly to detect lung nodules and cancer tissues. The difference-image processing proposed by Xu, which corresponds to the subtraction of a signal-enhanced image and a signal-suppressed image [6], is one method for suppressing the background structures. Shiraishi et al. proposed a removal segmentation method that first segmented the lung field and then filtered the background using the average radial-gradient technique [7]. The filter variable adaptively depends on the location of the region of interest (ROI) and the anatomical classification.

With the advancement of image processing studies, computer-aided diagnosis (CAD) studies and segmentation/classification of the medical image have been actively conducted to eliminate obstacles that interfere with diagnosis [8]. 

In biology fields, cell segmentation, and classification, the task of uniquely identifying each tumor cell in an image has shown significant progress [9]. Greenwald created the “Mesmer” method, a deep learning-enabled segmentation algorithm that performs nuclear and whole-cell segmentation in tissue imaging data. Mesmer, which achieves human-level performance for whole-cell segmentation, has enabled the automated extraction of key cellular features such as the subcellular localization of protein signals [10]

Staal proposed a 3D image segmentation framework that enables automatic segmentation and labeling of the entire rib on chest computed tomography (CT) [11]. For segmentation and labeling of complete ribs, the center of gravity of all ribs is calculated to decide centerlines. The ribs are labeled according to the calculated centerlines. This method was validated with 96.8% sensitivity, 97.8% specificity and 80% accuracy.

Similar to previous studies [6,7,8,9,10,11], rib segmentation can facilitate the accurate diagnosis of lesions. However, separating ribs on chest X-rays requires professional knowledge in medical anatomy. Additionally, it is difficult for experts to find all 12 pairs of ribs individually, and manually labeling them is laborious and time-consuming [12].

As a solution, rib segmentation using deep learning has been proposed. Segmentation using deep learning aids in obtaining useful information by performing rib suppression in CXRs, resulting in a more accurate diagnosis [13]. Due to the growing demand for CAD, deep learning has shown significant progress in medical imaging [14,15]. 

Arif et al. performed automatic segmentation of cervical spine X-ray images using a novel shape-aware deep segmentation network (Unet-S) and achieved a dice similarity coefficient (DSC) of 0.84 [16]. In 2020, Oliveira proposed a segmentation pipeline using a 3D model to aid in the rib segmentation of 2D X-ray images. It consisted of the MIP procedure for acquiring bone labels from CT scans, lung segmentation from CXR labels, and CoDAGAN for segmenting CXR ribs, achieving a maximum of 0.934 AUC [17].

Therefore, segmentation and image processing studies for rib diagnosis are well established. However, ribs overlap with various internal organs such as the lungs, heart, clavicle, and lower organs, making different characteristics for each rib region [18]. Using deep learning, these regional characteristics can be automatically extracted from the medical image [19], and we termed the image as having “regional characteristics”, which affect the deep learning process. Current studies have limitations in that deep learning has been conducted without considering the regional characteristics of ribs.

Hence, to minimize the difference in characteristics by rib regions, Wang proposed a multitask dense connection U-net (MDU-Net) that enabled multiple dense connections by combining Dense-Net with the U-Net model. The method separately divides the clavicle, anterior ribs, and posterior ribs, and each part passes multiple dense layers and then combines the segmentation results into one. This method succeeded in segmenting the ribs with 88% DSC [20].

Although rib images can be more clearly expressed by using image processing, filters, and 3D frames, it is necessary to verify whether a deep learning model delicately reflects the rib’s various features. However, there have been no reports verifying whether deep learning has completely reflected each region’s characteristics.

For complete segmentation of the entire rib, it is essential to consider the characteristics of each rib region. We hypothesized that deep learning, which did not reflect regional characteristics, causes regional performance differences. Therefore, this study quantitatively analyzed deep learning performance differences based on rib region to verify whether deep learning reflects the characteristics of each part. To compare segmentation performance, the rib image was divided based on the spine, clavicle, heart, and lower organs, which are characteristic indicators of the baseline chest X-ray. 

As a result of deep learning, the predicted ribs masks were divided horizontally into superior, middle, and inferior parts based on the clavicle and heart, and vertically into medial and lateral parts around the spine to determine which region specifically showed high performance (Figure 1).

Previous studies so far have focused on preprocessing rib images to provide reliable segmented images, and it is now necessary to verify how exactly deep learning reflects the characteristics of each region-specific feature.

Our study should provide essential pipelines for producing more strong rib segmentation algorithms in the future, and help experts reflect regional characteristics more accurately when applying deep learning to other medical images. 

## 2. Materials and Methods

### 2.1. Data Collection and Annotation

A total of 195 chest X-ray images formatted with digital imaging and communications in medicine (DICOM) were collected from Gil Hospital. The institutional review board of GUGMC approved this retrospective study and waived the requirement for informed patient consent (approval number: GBIRB-2019-337). Five experts who were trained to identify ribs for specialists demarcated the ROI, 12 pairs of ribs, on the images using ImageJ (NIH, Bethesda, MD, USA), and a specialist determined the final ROI after revision and modification.

### 2.2. Data Pre-Processing

For deep learning, resizing and zero-padding were applied to all the input images. The size was set to 1024×1024 pixels, with a value of 8 bits per pixel. Most deep learning methods show high performance when the number of training datasets is large. Therefore, to prevent overfitting during training and improve model performance, data augmentation was used for training data [21]. 

Three transformed images were produced from one training image by applying a rotation of approximately 10° and space augmentation or shifting translation of approximately 5%. Thus, 779 training images were generated with data augmentation, and the remaining 39 test images were used for verification. We used a five-fold cross-validation method to present the model’s performance quantitatively [22].

### 2.3. U-Net Architecture

A U-Net architecture, a generic deep learning solution for biomedical image segmentation based on an encoder–decoder structure, was used to extract the ribs’ image features [23,24,25,26]. As shown in Figure 2, the U-net has a U-shaped symmetrical form that includes a contracting path called an encoder to extract image features and an expansion path called a decoder to expand the feature map [27]. In particular, it is characterized by a U-net architecture with skip connections at the same depths to skip features from the contracting path to the expanding path to minimize spatial information lost during down-sampling [28].

The rectified linear unit (ReLu) function was used as the activation function, whereas the sigmoid function was used in the last convolutional layer. The hyperparameter was set to four batch sizes, 100 epochs, and a 0.001 learning rate. Additional information of the U-Net algorithm is presented in Table 1.

### 2.4. Criteria of Dividing Ribs

Original rib images (Figure 3b) were divided based on two criteria to compare the deep learning performance by region. The first criterion was to divide vertically into the medial and lateral sides along the lines of the spine (Figure 3a). The second criterion was to divide horizontally into the superior, middle, and inferior sides based on the clavicle, heart, and lower organs that consist of soft tissue (Figure 3b).

## 3. Results

To verify the performance of the rib segmentation model, 39 test images that were not used in training were used. The segmentation model performance was assessed by comparing the segmented images with the ground truth mask pixel-by-pixel. Three performance indicators were used: precision, recall, sensitivity, and DSC. True positive (TP) cases were those in which the rib was recognized as a rib, whereas false positive (FP) cases were those in which non-rib regions were recognized as the rib. Furthermore, true negatives (TNs) implied that non-rib regions were recognized as non-ribs, whereas false negatives (FNs) implied that ribs were incorrectly recognized as non-ribs. The precision, recall, sensitivity, and DSC were determined using Equations (1)–(4), respectively.
(1)$$\mathrm{Precision}=\frac{TP}{TP+FP}$$
(2)$$\mathrm{Recall}=\frac{TP}{TP+FN}$$
(3)$$\mathrm{Sensitivity}=\frac{TP}{TP+FN}$$
(4)$$\mathrm{DSC}=\frac{TP}{\left(TP+TN\right)+\left(FP+FN\right)}\times 2$$

### 3.1. Whole Ribs Segmentation Model Performance

Figure 4 depicts the segmented result images.

The segmentation result images were compared with the ground-truth images in the test dataset (39). On average, we observed a precision of 93.51 ± 0.04%, recall of 88.56 ± 0.86%, sensitivity of 88.14 ± 0.05%, and DSC of 89.65 ± 0.04 (Table 2).

### 3.2. Comparison Performance Difference between Medial and Lateral Ribs

Thirty-nine test sets were created, with entire rib images divided into medial and lateral regions based on the spine. The ground truth and predicted images were compared to show the differences between the two regions. This process is shown in Figure 5. 

On average, medial ribs had a precision of 89.756 ± 0.07%, a recall of 82.66 ± 0.82%, a sensitivity of 82.526 ± 0.08%, and a DSC of 84.756 ± 0.04%, while lateral ribs had a precision of 95.22 ± 0.03%, a recall of 90.89 ± 0.9%, a sensitivity of 91.03 ± 0.04%, and a DSC of 92.03 ± 0.03% (Table 3).

### 3.3. Comparison Performance Difference between Superior, Middle, and Inferior Ribs

Thirty-nine test sets were created, with all rib images divided into superior, middle, and inferior regions based on the clavicle, heart, and lower organs. The ground truth and predicted images were compared to show the differences between the three regions. This process is shown in Figure 6. 

On average, precision, recall, sensitivity, and dice similarity coefficient values in the superior ribs were 92.69 ± 0.04%, 88.92 ± 0.88%, 88.92 ± 0.05%, 89.94 ± 0.04%; 94.11 ± 0.04%, 90.31 ± 0.90%, 89.88 ± 0.05%, 91.01 ± 0.04% in the middle ribs; and 93.45 ± 0.04%, 83.53 ± 0.83%, 83.294 ± 0.1%, 85.8 ± 0.2% in the inferior ribs, respectively (Table 4).

## 4. Discussion and Conclusions

In this work, we postulated that deep learning training without considering the regional characteristics of ribs would cause differences in performance for each region. To verify this, we investigated whether significant differences occurred by comparing the deep learning performance of each region by dividing the ribs vertically and horizontally based on specific structures.

In the rib image, which was divided into medial and lateral ribs based on the spine, the performance was 7% higher in the lateral rib than in the medial rib. Additionally, rib images were divided into superior, middle, and inferior regions based on the clavicle, heart, and lower organs. The middle ribs showed the highest performance among them and presented an average of 6% higher performance compared with the inferior region, which showed the lowest performance. 

Therefore, it was confirmed that even if the same deep learning model was used, a performance difference of at least 6% was observed for each region. To comprehend these results, the characteristics of X-rays need to be considered.

The X-ray equipment commonly used in hospitals works on a principle in which particles of X-rays are irradiated to the body and transmitted, and the remaining X-rays are detected in the film [29]. Because the radiation density in our body is higher in the order of air, fat, organs, and bones, the lung area containing the most air appears dark, and the bone with the highest density appears bright in the X-ray image [30].

Therefore, in the image divided into medial and lateral based on the spine, it is difficult to distinguish between the spine and ribs because the density of the spine and ribs is similar. Moreover, because the boundary of the costal cartilage, which is set between the spine and ribs, is not clear to distinguish, it is considered that the medial region including the spine performed worse than the lateral region.

In rib segmentation, which is divided into superior, middle, and inferior, the inferior rib of the lower organ region, which contains more soft tissue, appears brighter, similar to other bones in the image. This makes the inferior region, which includes various organs, perform worse than the others.

Here, we used a normal chest X-ray image dataset at Gil Hospital. In the future, reinforcement learning with additional datasets of rib fractures and diseases such as complications of external fixation/traction or fracture union [31] can improve the developed model. 

We divided the entire ribs into two criteria based on the specific structures of the spine, clavicle, heart, and lower organs. In the future, comparing the performance by dividing the region of every 12 pairs of ribs will enable the discovery of new rib region segmentation criteria and serve as the foundation for studying rib fracture detection.

Figure 7 shows the error pixel, which makes false positives and false negatives. Therefore, to minimize these error pixels, we are currently further studying algorithms for rib labeling for dividing the region of every 12 pairs. It is expected that the revised algorithm will minimize FP and FN ratio by recognizing pixels more delicately in the future. We expect that our rib comparison study will improve the reliability of the segmentation method in digital radiography.

In this study, only U-Net was used among the CNN architectures, but regional performance comparisons with other CNN models are essential for the normalization regional performance differences [32]. We will conduct investigations to quantify and examine regional performance differences according to the architecture by using more advanced architectures such as the ResNet [33], GoogLeNet [34], and AlexNet [35] series in the future.

In conclusion, the chest X-ray deep learning model showed a 6%–7% difference in segmentation performance depending on the regional characteristics of the rib. This result shows that feature extraction technology that considers region specific characteristics is required in the deep learning process. 

Therefore, this study is meaningful in that it provides the importance of reflecting the characteristics of each region as well as presents the direction of erstwhile rib research. If feature extraction technology is developed through further research in the future, it is expected that the characteristics of each region can be expressed more precisely in medical images.

## Figures and Tables

**Figure 1 sensors-22-03143-f001:**
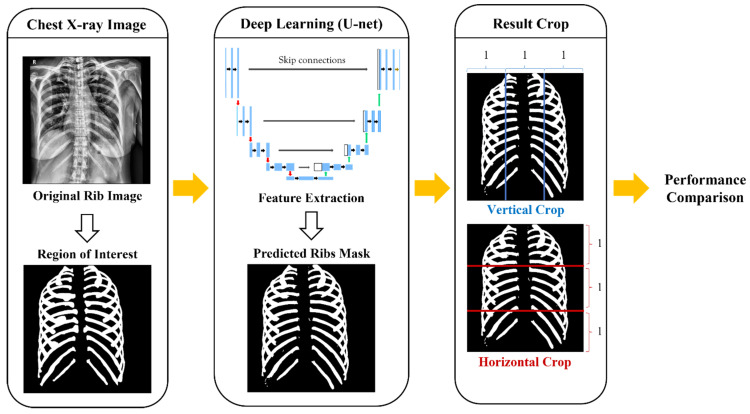
Flowchart that presents the total process of comparing region performance by dividing the predicted ribs’ mask.

**Figure 2 sensors-22-03143-f002:**
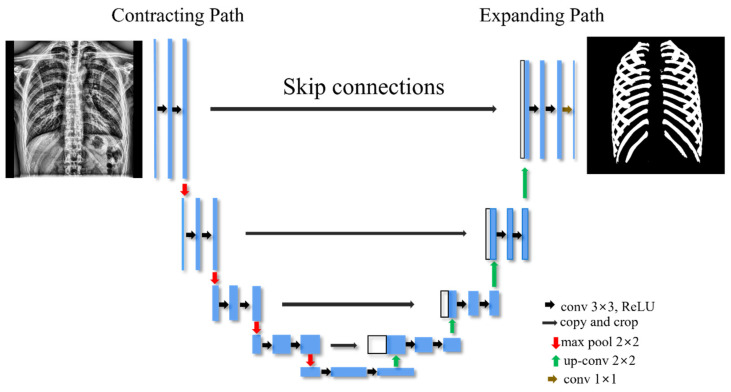
U-Net architecture for segmentation of ribs.

**Figure 3 sensors-22-03143-f003:**
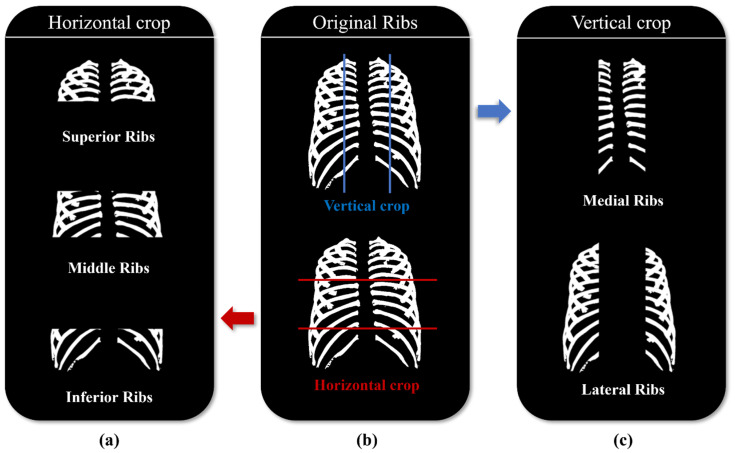
Criteria of dividing ribs: (**a**) Ribs vertically divide into medial and lateral of spine; (**b**) Original chest X-ray image; (**c**) Ribs horizontally divided into superior, middle, and inferior based on the clavicle, heart, and lower organs.

**Figure 4 sensors-22-03143-f004:**
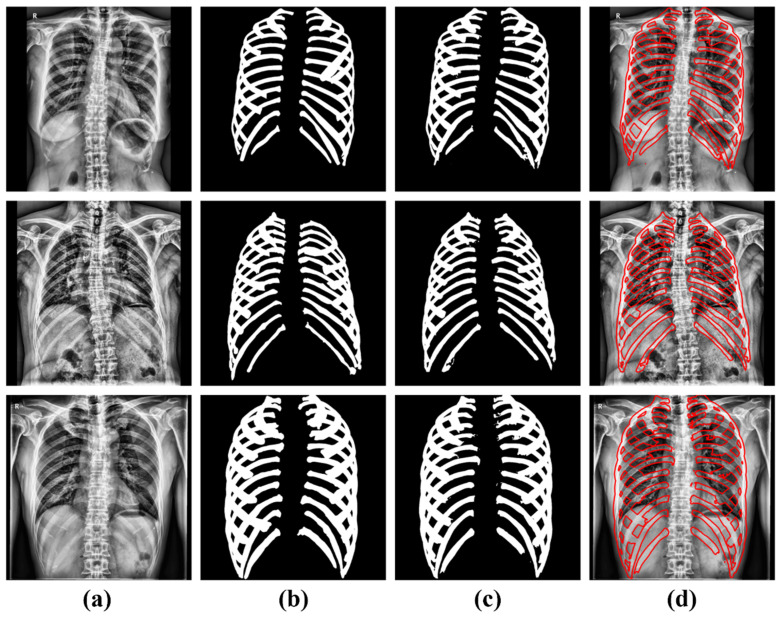
Segmentation model results. (**a**) Original images, (**b**) Ground truth images, (**c**) U-Net predicted images, and (**d**) Overlay images with (**a**,**c**).

**Figure 5 sensors-22-03143-f005:**
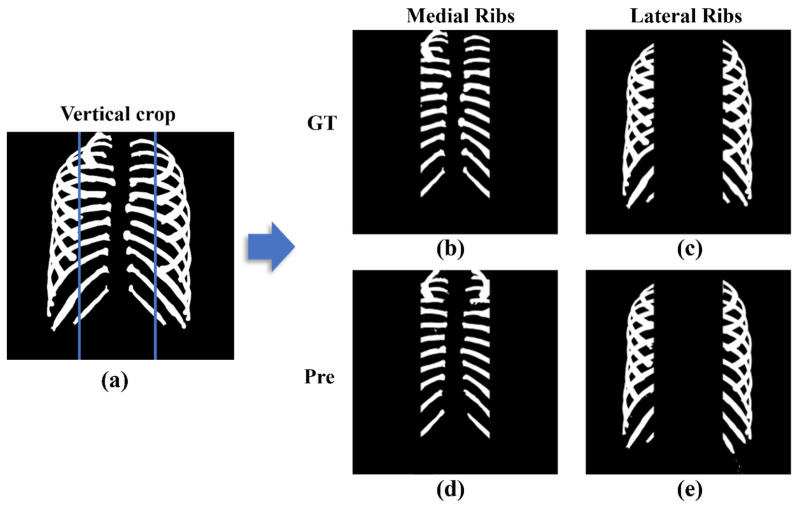
Vertical crop image divided into medial and lateral based on spine: (**a**) Entire rib image before cropping; (**b**) Ground truth images of medial ribs; (**c**) Ground truth images of lateral ribs; (**d**) Predicted images of medial ribs; and (**e**) Predicted images of lateral ribs.

**Figure 6 sensors-22-03143-f006:**
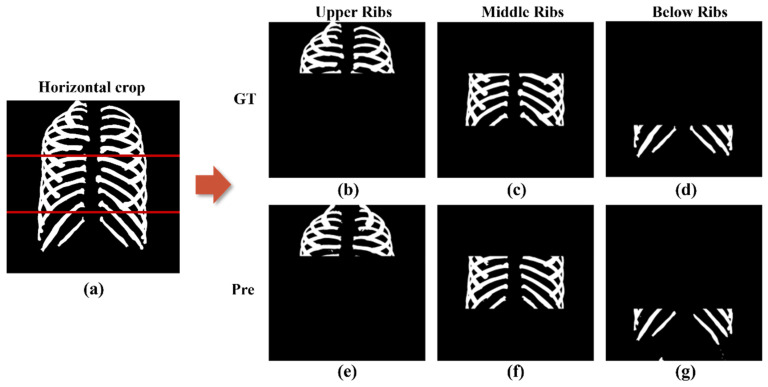
Horizontal crop image divided into superior, middle, and below based on the clavicle, heart, and lower organs: (**a**) Entire rib image before cropping; (**b**) Ground truth images of superior ribs; (**c**) Ground truth images of middle ribs; (**d**) Ground truth images of inferior ribs; (**e**) Predicted images of superior ribs; (**f**) Predicted images of middle ribs; and (**g**) Predicted images of inferior ribs.

**Figure 7 sensors-22-03143-f007:**
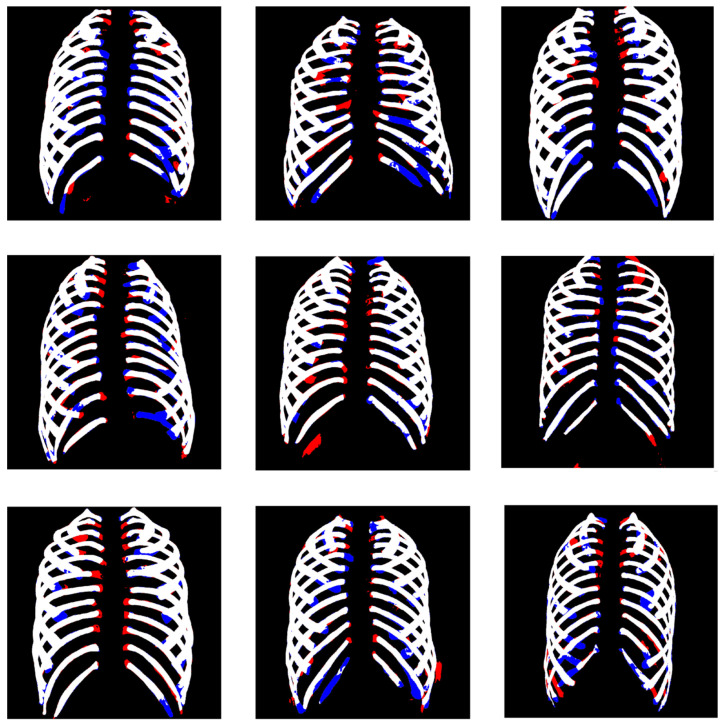
Segmentation predicted result images along with the false positives (FP: red color) and false negatives (FN: blue color).

**Table 1 sensors-22-03143-t001:** U-Net architecture and hyperparameters.

	Unit Level	Conv Layer	Filter	Activation Function	Output Size
Input					1024 × 1024 × 1
Encoding	Level 1	Conv 1	3 × 3/32	ReLu	1024 × 1024 ×32
		Conv 2	3 × 3/32		1024 × 1024 × 32
	Level 2	Conv 3	3 × 3/64	ReLu	512 × 512 × 64
		Conv 4	3 × 3/64		512 × 512 × 64
	Level 3	Conv 5	3 × 3/128	ReLu	256 × 256 × 128
		Conv 6	3 × 3/128		256 × 256 × 128
	Level 4	Conv 7	3 × 3/256	ReLu	128 × 128 × 256
		Conv 8	3 × 3/256		128 × 128 × 256
Bridge	Level 5	Conv 9	3 × 3/512	ReLu	64 × 64 × 512
		Conv 10	3 × 3/512		64 × 64 × 512
Decoding	Level 6	Conv 11	3 × 3/256	ReLu	128 × 128 × 256
		Conv 12	3 × 3/256		128 × 128 × 256
	Level 7	Conv 13	3 × 3/128	ReLu	256 × 256 × 128
		Conv 14	3 × 3/128		256 × 256 × 128
	Level 8	Conv 15	3 × 3/64	ReLu	512 × 512 × 64
		Conv 16	3 × 3/64		512 × 512 × 64
	Level 9	Conv 17	3 × 3/32	ReLu	1024 × 1024 × 32
		Conv 18	3 × 3/32		1024 × 1024 × 32
Output		Conv 19	1 × 1	Sigmoid	1024 × 1024 × 1

**Table 2 sensors-22-03143-t002:** Five-fold cross validation results of the trained segmentation model.

Folds	Precision (%)	Recall (%)	Sensitivity (%)	DSC (%)
Fold-1	92.06	87.54	87.17	88.95
Fold-2	92.72	87.61	87.62	89.1
Fold-3	97.74	88.44	87.41	89.74
Fold-4	97.76	90.14	89.13	91.08
Fold-5	87.25	89.04	89.39	89.39
Average	93.51 ± 0.04	88.56 ± 0.86	88.14 ± 0.05	89.65 ± 0.04

**Table 3 sensors-22-03143-t003:** Five-fold cross validation results of the medial and lateral ribs.

	Precision (%)	Recall (%)	Sensitivity (%)	DSC (%)
Medial Ribs	89.76 ± 0.07	82.66 ± 0.82	82.53 ± 0.08	84.76 ± 0.04
Lateral Ribs	95.22 ± 0.03	90.89 ± 0.9	91.03 ± 0.04	92.03 ± 0.03

**Table 4 sensors-22-03143-t004:** Five-fold cross validation results of the superior, middle, and inferior ribs.

	Precision (%)	Recall (%)	Sensitivity (%)	DSC (%)
Superior Ribs	92.69 ± 0.04	88.92 ± 0.88	88.92 ± 0.05	89.94 ± 0.04
Middle Ribs	94.11 ± 0.04	90.31 ± 0.90	89.88 ± 0.05	91.01 ± 0.04
Inferior Ribs	93.45 ± 0.04	83.53 ± 0.83	83.294 ± 0.1	85.8 ± 0.2

## Data Availability

The datasets generated and/or analyzed during the current study are not publicly available because the institutional review board did not grant permission to share patient data, but they are available from the corresponding author upon reasonable request.

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
