# Peer review of "Segmentation Performance Comparison Considering Regional Characteristics in Chest X-ray Using Deep Learning"

_sensors, 2022, doi:10.3390/s22093143_

Round 1
Reviewer 1 Report
Based on the in-depth analysis of the presented paper, a few significant issues have to be addressed:
- As the authors claim in the text, “this study, therefore, analyzes the differences in deep learning performance based on the rib region to verify whether deep learning reflects the characteristics of each part.” However, it is not clear from the paper what scientific goal this study pursues, as term “analyzes” is vague and must be implied in every research by default. Please, clearly indicate the scientific novelty of this work.
- It is also not understandable how dividing the ribs vertically and horizontally may explain differences in performance for each region. There might be a logical gap between the problem and the possible solution to it, as the primary objective seems not achievable. Please, carefully write down the hypothesis based on which the study is performed and clarify the method that is aimed to investigate the hypothesis.
- In addition to the remarks mentioned above, the authors are advised to describe the last paragraphs of Chapter 1 in more detail using graphic schemes of their approach.
- Terms “regional characteristics” and “regional features” of ribs are mentioned several times in the text; yet, it is unclear what features in the chest X-ray images are understood by these terms and how they are connected to investigated segmentation performance. Please, clarify these terms in the text.
- The authors claimed they had collected 195 chest X-ray images from Gil Hospital. However, a manually assembled dataset might be unevenly distributed with poor validity quality, which would negatively affect the generalized ability of the deep learning model. Thus, utilizing additional benchmarks for the rib segmentation task is strongly recommended:
1) VinDr-RibCXR (https://github.com/vinbigdata-medical/MIDL2021-VinDr-RibCXR);
2) RibSeg Dataset (https://zenodo.org/record/5336592#.YipDFHpBy5c);
3) or something else.
- Neural architecture selection and hyperparameter tuning are well-known cornerstones in deep learning. In this regard, it is not clear from the text why the authors chose the U-Net architecture over more advanced and robust ones, such as UNet++, ResUNet++, and their numerous modifications. The modern architectures might surpass slightly outdated U-Net from 2015 in the task of difference in segmentation performance. Please, specify the deep learning model with more details.
Reviewer 2 Report
In this paper, the deep neural network UNet is used to segment medical images to verify the experimental results. There are no obvious spelling errors in the paper, and the paper typesetting is very clean.
Major:
(1)Benchmark with other methods is necessary to show the good performace of the mentioned methods.
Reviewer 3 Report
In this manuscript, a segmentation performance comparison is presented by considering regional characteristics in chest X-ray images. The paper is relatively well written and technically sound, but the related work section provides a deficient analysis of recent approaches to biomedical image segmentation. Among the missing reference:
MoNuSAC2020: A multi-organ nuclei segmentation and classification challenge; IEEE Transactions on Medical Imaging, 2021.
* The structure of the manuscript could be improved. For example, a separate related work section should be added after the introduction.
* The manuscript contains several typos (please use a spell checker!!!) and grammatical errors.
* In the discussion section, the limitations of this work and how it can be improved should to be included.
Reviewer 4 Report
In this article, the authors present the analysis of deep learning performance differences based on rib region to verify whether deep learning reflects the characteristics of each part. To compare segmentation performance, the rib image was divided based on the spine, clavicle, heart, and lower organs, which are characteristic indicators of the baseline chest X-ray. They used the U-Net architecture to extract the ribs feature. The results show that the performance of the lateral rib is 7% higher than in the medial rib in the vertical division. While the middle ribs showed the highest performance (6%) among superior, middle, and inferior regions in the horizontal division. The overall article is well written but it requires some improvements that are mentioned below.
1) In the abstract, please include some details about the dataset that you have used. Please include your results with other state-of-the-art models used for segmentation. Only U-net is not enough.
2) A separate section for the contributions and related work must be included.
3) At the end end of related work, make a comparison table that should highlight the strengths and weaknesses of the proposed and previous methods.
4) In the introduction section, please include an overall flow chart to explain your proposed method.
5) In results, please include the results of segmentation along with the false positives and false negatives. Assign different colors for each error class.
Round 2
Reviewer 1 Report
Dear authors, in general, your answers to my remarks are correct and make the corrected original appearance of the paper more understandable.
For my opinion, in this state, this paper is recommended to be accepted in its current form
Reviewer 4 Report
Most of my comments are addressed. I recommend acceptance of this article.